# Guanfacine Normalizes the Overexpression of Presynaptic α-2A Adrenoceptor Signaling and Ameliorates Neuropathic Pain in a Chronic Animal Model of Type 1 Diabetes

**DOI:** 10.3390/pharmaceutics14102146

**Published:** 2022-10-10

**Authors:** Neha Munawar, Joelle Nader, Najat H. Khadadah, Ashraf Al Madhoun, Waleed Al-Ali, Linu A. Varghese, Willias Masocha, Fahd Al-Mulla, Milad S. Bitar

**Affiliations:** 1Department of Pharmacology and Toxicology, Faculty of Medicine, Kuwait University, Al-Jabriya 046302, Kuwait; 2Department of Mathematics and Natural Sciences, American University of Kuwait, Salmiya 20002, Kuwait; 3Department of Genetics and Bioinformatics, Dasman Diabetes Institute, Dasman 15400, Kuwait; 4Department of Animal and Imaging Core Facilities, Dasman Diabetes Institute, Dasman 15400, Kuwait; 5Department of Pathology, Faculty of Medicine, Kuwait University, Al-Jabriya 046302, Kuwait; 6Department of Pharmacology and Therapeutics, College of Pharmacy, Kuwait University, Al-Jabriya 046302, Kuwait

**Keywords:** diabetic neuropathic pain, thermal hyperalgesia, cold allodynia, mechanical allodynia, alpha-2A adrenoceptors, guanfacine

## Abstract

Background: Diabetes is associated with several complications, including neuropathic pain, which is difficult to manage with currently available drugs. Descending noradrenergic neurons possess antinociceptive activity; however, their involvement in diabetic neuropathic pain remains to be explored. Methods: To infer the regulatory role of this system, we examined as a function of diabetes, the expression and localization of alpha-2A adrenoceptors (α2-AR) in the dorsal root ganglia and key regions of the central nervous system, including pons and lumbar segment of the spinal cord using qRT-PCR, Western blotting, and immunofluorescence-based techniques. Results: The data revealed that presynaptic synaptosomal-associated protein-25 labeled α2-AR in the central and peripheral nervous system of streptozotocin diabetic rats was upregulated both at the mRNA and protein levels. Interestingly, the levels of postsynaptic density protein-95 labeled postsynaptic neuronal α2-AR remained unaltered as a function of diabetes. These biochemical abnormalities in the noradrenergic system of diabetic animals were associated with increased pain sensitivity as typified by the presence of thermal hyperalgesia and cold/mechanical allodynia. The pain-related behaviors were assessed using Hargreaves apparatus, cold-plate and dynamic plantar aesthesiometer. Chronically administered guanfacine, a selective α2-AR agonist, to diabetic animals downregulated the upregulation of neuronal presynaptic α2-AR and ameliorated the hyperalgesia and the cold/mechanical allodynia in these animals. Conclusion: Together, these findings demonstrate that guanfacine may function as a potent analgesic and highlight α2-AR, a key component of the descending neuronal autoinhibitory pathway, as a potential therapeutic target in the treatment of diabetic neuropathic pain.

## 1. Introduction

Diabetes mellitus is a life-threatening chronic condition affecting approximately 451 million people worldwide. These figures may rise to 693 million by 2045 [1]. Diabetic neuropathy is considered one of the most frequent clinical complications of diabetes [2]. In Type 1 Diabetes Mellitus (T1DM), the prevalence of diabetic neuropathy ranges from 7 to 34.2% [3,4,5,6]. Approximately 20 to 30% of diabetic patients with neuropathy suffer from neuropathic pain [7,8,9]. Diabetic neuropathic pain is characterized by hyperalgesia, allodynia, or spontaneous sensations, including electric shock-like, pins and needles, shooting, lancinating, or burning [10,11,12]. Diabetic neuropathic pain impairs daily living activities, and thus, has a negative effect on patients’ quality of life (QOL) [12].

The pathogenesis of diabetic neuropathic pain is complex and multifactorial. Various theories have been proposed to explain the pathogenesis of pain associated with diabetic neuropathy, including reactive oxygen species-induced oxidative stress, metabolic and autoimmune disorders accompanied by glial cell activation, changes in sodium, calcium, and potassium channels expression, changes in the blood vessels that supply the peripheral nerves, dysregulation of the noradrenergic and/or serotonergic descending pathways, and peripheral and central sensitization mechanisms [13]. Management of diabetic neuropathic pain remains challenging. Treatment is mainly symptomatic, although proper glucose control is the primary mainstay for diabetic neuropathy treatment and to control potential modifiable risk factors. At present, drug classes that have received regulatory approval for the management of painful diabetic neuropathy by the United States Food and Drug Administration (FDA) include anticonvulsants, tricyclic antidepressants, serotonin, and noradrenaline reuptake inhibitors (SNRIs), and opioids [12,14]. Based on their efficacy, the pharmacological treatments can reduce pain and improve the patient’s QOL; however, they are not sufficient to prevent neuropathy development, or reverse existing disease and associated pain. Furthermore, combination of different drug classes accounts for only 50% of pain management in less than one-third of patients, indicating significant clinical practice challenges [15]. Therefore, there is a need for novel therapeutic drugs that can effectively target the underlying pathophysiology and reverse the defects of diabetic neuropathy.

Previous studies have shown that the noradrenergic system is altered during diabetes. Streptozotocin (STZ)-induced diabetic rats are more sensitive to mechanical stimuli in comparison to non-diabetic rats [16], have lower tyrosine hydroxylase activity and notable reduction in lumbospinal noradrenaline (NA) release (3-Methoxy-4-hydroxyphenyl-glycol (MHPG)/NA ratio) compared to controls [16,17,18]. Interestingly, alpha-2 adrenoceptor (α2-AR) agonists, tricyclic antidepressants, and NA reuptake inhibitors (NRIs) were shown to be effective in ameliorating diabetic neuropathic pain or neuropathic pain due to ligated nerves [19,20,21,22].

Although, α2-ARs may be regarded as therapeutic targets, there is a need to evaluate potential clinical candidate drugs with a high degree of selectivity and efficacy. α2-ARs are predominantly involved in neurotransmission [23,24]. A significant proportion of α2-ARs is located in the locus coeruleus [25], and their activation mediates antinociception [26]. In the central nervous system, α2-ARs are distributed abundantly in the locus coeruleus, cerebral cortex, brain stem, hypothalamus, septum, amygdala, and hippocampus [23]. Additionally, α2-ARs are also present across the dorsal and ventral horn and on the central terminals of nociceptive nerve fibers at the level of the spinal cord [25]. They also function as auto-receptors on pre-synaptic neurons [27].

Guanfacine is a selective α2-AR agonist. In contrast, other drugs such as clonidine target both α2-ARs and imidazoline receptors [28]. Administration of guanfacine has been reported to produce dose-dependent antinociception in mouse models of formalin-induced inflammatory colonic pain [29]. Guanfacine extended-release tablets (Intuniv) are FDA-approved drugs for the treatment of attention deficit hyperactivity disorder in children aged between six and seventeen years [30,31,32]. Oral guanfacine monotherapy was also found to improve chronic treatment-refractory cough in the case of a 58-year-old woman who had an eight-year history of neurogenic symptoms [33]. Currently, the efficacy of guanfacine in combination with lidocaine versus lidocaine alone is being evaluated in 30 patients with painful trigeminal neuropathy in a double-blind randomized clinical trial conducted at the Vanderbilt University Medical Center in Tennessee, United States (ClinicalTrials.gov Identifier: NCT03865940). Guanfacine has shown promising results in different models; however, the use of this compound has not yet been investigated in an experimental chronic model of T1DM-induced neuropathic pain. Therefore, the objective of this study was to investigate the hypothesis that the development of neuropathic pain stems from a defect in the central noradrenergic system activity in T1DM and chronic administration of guanfacine reverses the defect of the noradrenergic system and ameliorates neuropathic pain.

## 2. Materials and Methods

### 2.1. Animals

Male Wistar rats (150–200 g) were bred and supplied by the Animal Resources Center at the Health Sciences Center (HSC), Kuwait University, Kuwait. All experiments were approved by the Ethical Committee for the use of Laboratory Animals in Teaching and in Research, HSC, Kuwait University, Kuwait, in accordance with the guidelines of the Animal Research: Reporting of In vivo Experiments (ARRIVE). All rats were housed in open top cages on a 12-h light/dark cycle at a temperature of 24 ± 1 °C with access to food and water ad libitum. All experiments were performed at the same period of the day (8:00 a.m. to 16:00 p.m.) to exclude diurnal variations in pharmacological effects.

### 2.2. Induction of Diabetes

Rats were randomly divided into vehicle (20 rats, non-diabetic) and treatment (50 rats, diabetic) groups. After 16 h of fasting and free access to water, rats were weighed, and rats in the diabetic group were injected with STZ (Sigma-Aldrich, St Louis, MO, USA). Immediately before use, STZ was dissolved in 0.05 M sodium citrate buffer, pH 4.5, and was administered at a dose of 55 mg/Kg once intraperitoneally to induce T1DM [34]. Rats in the control were injected with 0.05 M sodium citrate buffer alone. An hour after injections, rats were allowed free access to food and water. STZ-injected animals were monitored daily for the development of T1DM symptoms. To maintain hygiene and avoid infections, beddings were changed daily.

### 2.3. Measurement of Blood Glucose, Body Weight, Food Consumption and Water Intake

Three days after injections, the diabetic status of the rats was confirmed by measuring fasting blood glucose concentrations. Blood samples were taken from the tail vein and glucose was measured using a glucometer (Accu-Chek Performa, Roche, Basel, Switzerland) as previously described [35]. Four weeks after injections, an intraperitoneal glucose tolerance test (IP-GTT) was performed as previously described [36,37]. Following an overnight fast, blood glucose values of control and diabetic rats were measured at baseline (0 time-point). Rats in each group were then administered 2 g/Kg body weight of glucose (20% glucose in sterile water) intraperitoneally. Glucose values were measured at 15, 30, 60, 90 and 120 min.

Body weights of buffer-treated controls and diabetic rats were recorded weekly throughout the experimental study. For recording consumption of food and water intake, the rats were fasted overnight and allowed access to 100 g feed and 100 mL water for two hours. The food consumption was determined by calculating the difference between the pre-weighed feed (100 g) and the weight of the feed remaining per body weight of each rat. Similarly, water intake was recorded by measuring the quantity of water consumed (100 mL-volume of water remaining in the bottle) per body weight of each rat.

### 2.4. Drugs Administration

To study the acute effects of guanfacine (GF, TCI America, Portland, OR, USA), GF was administered at doses of 0.15, 0.3, 0.6, 1.2 and 5 mg/Kg. A dose 0.6 mg/Kg was chosen as described previously [38]. Further, we used the other doses to plot a dose response curve. Guanfacine was dissolved in deionized water and was freshly prepared before administration. After the development of T1DM, STZ-treated rats (referred to as T1DM and/or diabetic rats across the manuscript) were randomly divided into six groups: I: T1DM; II: T1DM+GF-0.15; III: T1DM+GF-0.3; IV: T1DM+GF-0.6; V: T1DM+GF-1.2; and VI: T1DM+GF-5. Similarly, the normal control rats (referred to as NC and/or control rats across the manuscript) were injected with three doses of GF (0.6, 1.2, and 5 mg/Kg) and were randomly grouped into four groups: I: NC; II: NC+GF-0.6; III: NC+GF-1.2; and IV: NC+GF-5.

To evaluate the effects of the α2-AR antagonist, yohimbine (YOH, Sigma-Aldrich, St Louis, MO, USA), animals were pre-treated with 1 mg/Kg [39]. To study the role of YOH on the anti-hyperalgesic and anti-allodynic effects of GF, rats were randomly divided into four groups: I: T1DM; II: T1DM+YOH-1; III: T1DM+GF-0.6; and IV: T1DM+GF-0.6 +YOH-1. As described previously, the drug compounds were prepared freshly on the day of the experiment and administered intraperitoneally to the rats. YOH was administered 15 min before the administration of guanfacine for the thermal (heat/cold) and mechanical behavioral tests [40].

To compare the anti-hyperalgesic and anti-allodynic effects of clonidine versus guanfacine, both clonidine and guanfacine in separate experiments were administered daily for a period of two weeks after the development of thermal hyperalgesia and cold/mechanical allodynia (Appendix A shows a schematic diagram for the experimental procedure). The reaction latency times to thermal (heat/cold) and withdrawal thresholds to mechanical stimuli were measured at weeks 8, 9, 10, 11 and 12. During weeks 10–12, rats were not administered the drugs. The rats were divided into four groups for guanfacine experiment, I: NC; II: NC+GF-0.6; III: T1DM, and IV: T1DM+GF-0.6. Similarly, for the clonidine experiment, rats were divided into four groups: I: NC; II: NC+Clonidine-0.1; III: T1DM, and IV: T1DM+Clonidine-0.1.

### 2.5. Assessment of Thermal Hyperalgesia, Cold Allodynia and Mechanical Allodynia

Development of thermal hyperalgesia, cold allodynia, and mechanical allodynia in response to thermal (heat/cold) and mechanical stimuli were assessed at baseline and on a weekly basis until development of nociceptive behavior towards each stimulus, as described previously [41,42,43], was detected. To evaluate the anti-hyperalgesic and anti-allodynic responses after clonidine or guanfacine administration, pretreatment values were recorded prior to chronic administration of the agonists for a period of two weeks. Pain transmission tests were performed at weekly intervals for two weeks of guanfacine administration followed by two weeks of drug withdrawal.

For the acute effect and time/dose-response curves of guanfacine, rats were administered separate intraperitoneal doses of guanfacine (0.15, 0.3, 0.6, 1.2 and 5 mg/Kg) at the same time and pain threshold in response to thermal (heat/cold) and mechanical stimuli were measured at different time-points (0, 15, 30, 60, 90 and 120 min).

Thermal hyperalgesia was assessed using the Hargreaves test (Ugo Basile, Italy) to measure paw withdrawal reaction latency in response to infrared heat [41]. The Hargreaves test comprises a controller, infrared radiant (IR) emitter, framed glass panel and three compartments with six spaces to enclose the rats. Briefly, each rat was individually placed in the compartment and allowed to habituate for 30 min. A portable infrared heat source was positioned underneath the plantar surface of the hind paw. The time taken for the paw to withdraw from the IR heat (IR intensity: 50 units) was recorded as the paw withdrawal latency. The intensity of the IR emitter was graded from 0 to 99. We chose 50 to obtain a response within the cut-off time (30 s) to avoid damage to the paws. The test was repeated twice per paw in each rat with a 30 s interval, and the average of the four paw withdrawal latencies for each rat was recorded.

Cold allodynia was performed as described previously [43] with some modifications. Rats were individually placed on a cold plate designed in our laboratory with the temperature adjusted to 4 ± 0.1 °C. The cold-plate device was comprised of a circular cold-plate enclosed in cylindrical chamber. The chamber was covered and placed in a plexiglass square box consisting of ice. On the experimental day, the reaction time in seconds between the placement of the rat on the cold-plate and the first sign of licking or jumping was recorded. To avoid damage to the paws, an arbitrary cut-off time of 70 s was maintained.

Mechanical allodynia was measured using the dynamic plantar aesthesiometer (Ugo Basile, Italy) as previously described [42]. Briefly, rats were first trained for three to five days to acclimatize to the aesthesiometer, and baseline readings were recorded for a period of three days. On experimental day, rats were left to habituate for about 30 min inside plastic enclosures on top of a mesh platform before activating a microprocessor. The microprocessor was programmed to automatically lift a metal filament that exerts a linearly increasing force (2.5 g/s) from below the mesh platform to the plantar surface of the hind paw with the help of an adjustable angled mirror. The filament was automatically withdrawn when the rat withdrew its paw or at a cut-off force of 50 g. Withdrawal thresholds in response to the mechanical stimulus were automatically recorded in grams. Each rat was tested three times, allowing at least 30 s between each measurement.

### 2.6. Tissue Isolation

Rats were anesthetized with a combination of xylazine (Interchemie Werken, Holland, Netherlands) and ketamine (Dutch Farm International, Holland, The Netherlands), and sacrificed by a trained veterinarian, as described previously [18]. Pons tissue at the brain stem region, the lumbar segment of the spinal cord (LSSC, segment L1-L4) and dorsal root ganglia of the spinal cord (DRG) were isolated, snap frozen in liquid nitrogen, powdered with a motor and pestle, and stored at −80 °C [18].

### 2.7. RNA Extraction, cDNA Synthesis and qRT-PCR Reactions

Total RNA was extracted from the fresh frozen powdered tissues using the TRIzol reagent (Invitrogen, Carlsbad, CA, USA) as described by the manufacturer. The first strand cDNA was synthesized from 1 µg RNA using High-Capacity cDNA Reverse Transcription kit (Applied Biosystems, CA, USA). Real-time qRT-PCR was performed as described [44]. Then, PCR reactions were performed on an ABI Prism 7500 Fast Real-Time PCR System (Applied Biosystems, CA, USA). Each cycle involved denaturation (15 s at 95°C), annealing/extension (1 min at 60 °C) after UDG (2 min at 50 °C) and AmpliTaq gold enzyme (10 min at 95 °C) activation, as previously described [45]. The primer sequences for beta-actin were forward, 5′- CCGCGAGTACAACCTTCTTG -3′ and reverse, 5′- GCAGCGATATCGTCAATCCAT -3′; and for α2-AR were forward, 5′- GGTAAGGTGTGGTGCGAGAT -3′ and reverse, 5′- CAGCGCCCTTCTTCTCTATG -3′. Relative gene expression to control was calculated using comparative Ct method, as previously described [46]. Results were normalized to beta-actin, and averages ± SEM are shown expressed relative to controls as indicated [47].

### 2.8. Western Blot, Immunohistochemistry, and Immunofluorescence of Tissue Sections

Western blot analysis was performed as described previously [48]. The neuronal tissues were homogenized in 1X cell lysis buffer (Cell Signaling, Danvers, MA, USA) containing 1 mM phenylmethylsulfonyl fluoride (Sigma-Aldrich, St Louis, MO, USA). Total proteins were quantified using the Pierce BCA Protein Assay Kit (Thermo Fisher Scientific) and equal amounts of protein were resolved on 8–12% polyacrylamide gels and transferred to polyvinylidene fluoride membranes (EMD Millipore Corporation, Billerica, MA, USA). After blocking, the membranes were blotted with the corresponding primary and horseradish peroxidase-linked secondary antibodies (Table 1). Proteins were visualized using Amersham ECL Prime Western Blotting Detection Reagent (GE Healthcare Life Sciences, Chicago, IL, USA). Images were captured using the VersaDoc™ MP 5000 imaging system (Bio-Rad Laboratories, Hercules, CA, USA).

Tissues were fixed with 4% formalin, embedded in paraffin, and sectioned (5 μm in thickness). Following de-paraffinization and rehydration, tissue sections were subjected to immunohistochemistry staining as described previously [49]. Briefly, tissue sections were rinsed with distilled water, placed in sodium citrate buffer (pH 6.0) and heated for 20 min in a microwave (BEC, Shanghai, China) for antigen retrieval. Following incubation at room temperature for 30 min, the sections were rinsed with distilled water. Endogenous peroxidase activity was blocked with 3% hydrogen peroxide solution for ten minutes to quench endogenous peroxidase activity. The sections were then rinsed with distilled water and phosphate-buffered saline (PBS). Non-specific antigen-antibody reactions were then inhibited by treatment with protein block (Agilent, Santa Clara, CA, USA) for 15 min at room temperature. After rinsing with PBS, the sections were incubated with primary antibody overnight and then with the secondary antibody horseradish peroxidase anti-rabbit IgG. For chromogen reactions, 3,3′-diaminobenzidine was used at room temperature within two minutes. The sections were then rinsed in deionized water, running tap water, and counterstained in filtered, acidified Mayer’s Hematoxylin stain. Following staining with hematoxylin, the sections were immersed in ascending ethanol concentrations and xylene. Immunofluorescence was performed as described previously [50]. Briefly, fixed tissues sections were incubated overnight with appropriate primary antibodies and then with fluorescence secondary antibodies (Table 1). Finally, images were taken at Zeiss Axiovert 200 fluorescence microscope equipped with a cooled CCD camera (Carl Zeiss Microscopy GmbH, Jena, Germany) and linked to an IBAS 2000 image analyzer for capturing and analyzing pictures.

### 2.9. Synaptosome Extraction and Isolation of Pre-Synaptic and Post-Synaptic Fractions

Synaptosomes extraction was performed using a discontinuous sucrose gradient method, as described previously [51]. Briefly, LSSC were homogenized in isolation buffer containing 0.32 M sucrose, 0.1 mM calcium chloride, and 0.1 mM magnesium chloride, pH 7.4. The homogenate solution was mixed with 6 mL of 2 M sucrose and 2.5 mL of 0.1 mM calcium chloride. The homogenate was centrifuged at 100,000× *g* for three hours at 4 °C. The generated white ring corresponding to the synaptosome fraction (1.25 M and 1 M interphase) was collected. The synaptosomes were diluted and re-centrifuged for 30 min at 15,000× *g* at 4 °C, and then the pellet was resuspended in 1 mL isolation buffer.

To isolate the pre-synaptic and post-synaptic fractions, synaptosome fraction suspension was diluted with 5 mL of 0.1 mM calcium chloride and 5 mL of ice-cold 2X solubilization buffer (pH 6). The fractions were incubated with high agitation in a beaker on ice for 50 min. Then, the solution was centrifuged for 30 min at 40,000× *g* at 4 °C. Pellets containing the pre-synaptic and post-synaptic fractions were washed with 1X solubilization buffer (pH 6), resuspended with 1X solubilization buffer (pH 8) and incubated in a beaker on ice for 50 min under high agitation. The solution was centrifuged at 40,000× *g* for 30 min at 4 °C. The pellet corresponding to the post-synaptic fraction was resuspended in 5% sodium dodecyl sulfate (SDS) buffer. The supernatant corresponding to the pre-synaptic fraction was concentrated using Millipore ultrafiltration centrifugal tubes (Sigma-Aldrich, St Louis, MO, USA), precipitated with 5 volumes of pre-chilled acetone overnight at −20 °C and centrifuged at 18,000× *g* for 30 min. The pre-synaptic pellet was dried and resuspended in 5% SDS buffer.

### 2.10. Statistical Analyses

Statistical analyses were performed using an unpaired Student’s *t*-test and one-way analysis of variance (ANOVA), followed by Dunnett’s multiple comparison post-tests or two-way repeated-measures ANOVA followed by Bonferroni’s multiple comparison post-tests using GraphPad Prism software (version 7.0) [52]. The differences were considered significant at *p* < 0.05. The results in the text and figures are expressed as the means ± SEM.

## 3. Results

### 3.1. Biochemical, Physical and Behavioral Characteristics of Control and T1DM Rats

At day 3 after treatment, STZ injected rats developed hyperglycemia as indicated by the elevation in the levels of blood glucose, >20 mmol/mL, relative to the control groups, <5 mmol/mL (Figure 1A). Additionally, as shown in the IP-GTT test, glucose levels were significantly higher in diabetic rats in comparison to control animals at all tested time-points, a trend paralleled by calculated area under the curve (AUC) values (Figure 1B,C). The glucose values peaked at 30 min for both control and diabetic rats. However, when compared to baseline values, diabetic rats showed significant increases in glucose values starting at 15 min until 120 min, whilst control rats showed increased glucose values at 15, 30, 60 and 90 min but not at 120 min (Figure 1B).

The observed phenotype was sustained for eight weeks after STZ treatment. Unlike the control group, the T1DM animals had a statistically significant gradual reduction in their body weight (almost >20%) during the study period (Figure 1D). Furthermore, the diabetic rats exhibited significant increase in food consumption (53 ± 5 mg/body weight/2hrs) in comparison to vehicle-treated rats (25 ± 2 mg/body weight/2hrs) at the week 8 time-point (Figure 1E, *p* ≤ 0.05). Water intake was also elevated in the diabetic rats (0.139 ± 0.006 mL/body weight/2hrs) relative to the corresponding controls (0.026 ± 0.004 mL/body weight/2 h) at the week 8 time-point (Figure 1F, *p* ≤ 0.05).

Diabetes resulted in a significant reduction in pain threshold to thermal (heat/cold) and mechanical stimuli starting week 1, in comparison to the corresponding baseline values and control animals (Figure 1G–I, *p* ≤ 0.05). At week 8, reaction latencies in diabetic rats to heat stimuli dropped to 4.3 ± 0.56 s when compared to control rats, 19.41 ± 0.81 s (Figure 1G). Similarly, their reaction latencies to cold stimuli were reduced to 21.46 ± 1.39 s compared to 60.43 ± 2.11 s for the controls (Figure 1H). The same trend was also observed for the withdrawal thresholds in response to mechanical stimuli; the responses of diabetic rats were reduced by three-fold relative to controls (Figure 1I).

### 3.2. Time and Dose Response Curves of Administered Guanfacine in Control and T1DM Rats

In control rats, intraperitoneal administration of guanfacine at various doses (0.6, 1.2 or 5 mg/Kg), did not alter pain threshold to the thermal (heat/cold) or mechanical stimuli (Figure 2A–C), whereas diabetic rats treated with guanfacine showed notable changes in responses to the tested stimuli. In these animals, guanfacine treatment produced significant effects 30 min after administration that persisted for up to 120 min (Figure 2D–F, *p* ≤ 0.05). Furthermore, increasing concentrations of guanfacine did not alter the pain responses of the control rats to the different insults (Figure 2G–I). On the other hand, the diabetic animals’ pain response was notable, and the effective concentration, EC50, for guanfacine was 0.36 mg/Kg in the thermal (heat/cold) test (Figure 2J,K), whereas EC50 = 0.34 mg/Kg was observed in the mechanical test at 30 min after administration (Figure 2L) in diabetic rats.

Using the diabetic rats, we studied the effects of the yohimbine, α2-AR antagonist. As shown in Figure 2M-O, treatment with yohimbine alone did not alter the pain threshold to thermal (heat/cold) and mechanical stimuli when compared to vehicle-treated rats and baseline values. Nevertheless, yohimbine treatment was sufficient to inhibit the anti-hyperalgesic and anti-allodynic actions of guanfacine. Treatment of diabetic rats with yohimbine followed by guanfacine resulted in a statistically significant reduction (40–50%) in the animals’ responses to the various tested stimuli when compared to guanfacine alone, which in turn showed significant elevation in the animals’ responses as observed in Figure 2M–O.

### 3.3. Persistent Guanfacine Analgesic Effects Following Two Weeks of Withdrawal

Chronic treatment with guanfacine significantly elevated pain threshold to thermal (heat/cold) and mechanical stimuli in comparison to corresponding vehicle-treated rats. Interestingly, the influence of guanfacine on the pain threshold of the studied stimuli was sustained after a two-week period of drug withdrawal (Figure 3A–C).

In contrast to control animals, which did not show any responses to drug, the diabetic rats’ reaction latency times to thermal (heat) stimuli was elevated by 2.75 and 2.2-fold relative to the baseline and to untreated animals, respectively, even after drug excision (Figure 3A). Similarly, guanfacine treatment improved the responses to thermal (cold) and mechanical stimuli specifically in diabetic rats by 1.5 to 2-fold, respectively, under the applied experimental conditions (Figure 3B,C). Together, our data suggests that guanfacine treatment is sufficient to reverse diabetic animal response to thermal (heat/cold) and mechanical stimuli, most likely by improving several neuronal signals that are yet to be fully demonstrated.

### 3.4. Chronically Administered Guanfacine Halts Diabetes-Induced Neuronal Degeneration

To demonstrate the effect of diabetes and the influence of guanfacine treatment on neuronal tissues, we isolated LSSC and DRG tissues and stained them using Hematoxylin and Eosin (H&E) protocols. As observed in Figure 4, unlike in control rats, the LSSC tissue in diabetic animals showed ischemic neuronal cells, Nissl body loss and pyknosis, eosinophilic cytoplasm, and glial cells with cellular edema (Figure 4A,B, also for details Appendix A). On the other hand, treatment with guanfacine resulted in a notable improvement of the LSSC, yet it was not completely reversed to the level of the vehicle-treated control group (Figure 4C).

H&E staining of the DRG from diabetic rats revealed a notable satellitosis and neuronophagia with basophilic, vacuolar-like defects and pigmentary inclusions (Figure 4E), which was not observed in DRG isolated from the control group (Figure 4D). Alternatively, H&E staining of the DRG from diabetic rats treated with guanfacine showed a significant improvement of the tissue, as observed in Figure 4F.

### 3.5. Chronically Administered Guanfacine Reverses Diabetes-Induced Upregulation of Neuronal α2-AR

At the molecular level, statistically significant elevated levels of the α*2-AR* transcripts were observed in pons, LSSC and DRG tissues extracted from the diabetic rats in comparison to those isolated from the controls (Figure 5A, *p* ≤ 0.05). Remarkably, treatment with guanfacine was sufficient to normalize the levels of α*2-AR* mRNA in diabetic animals in all the studied neuronal tissues (Figure 5A) to levels comparable to that in control animals. In accordance with the qRT-PCR results, Western blot analyses also revealed a significant increase in the levels of α2-AR proteins in pons, LSSC and DRG obtained from diabetic rats compared to vehicle-injected controls (Figure 5B), which was significantly reduced after guanfacine treatment in all studied neuronal tissues (Figure 5B).

We further tested α2-AR protein levels using immunohistochemical staining. As shown in Figure 5C, α2-AR proteins were detected in abundance in cells within the LSSC tissue from diabetic rats as compared to those from controls, which was eventually reduced in response to guanfacine administration (Figure 5C). A similar trend for α2-AR protein expression was also noticed in DRG tissues. Together, these data indicate that elevated levels of α2-AR are associated with diabetes and are dramatically reduced in response to guanfacine treatment. Although the mechanism by which guanfacine reduces α2-AR levels was not elucidated in the current study, it is possible that guanfacine may alter the half-life or reduce the transcriptional rate of α2-AR.

### 3.6. Expression of α2-AR in the Pre-Synaptic and Post-Synaptic Fractions of the Spinal Cord Lumbar Region

Synaptosomal associated protein-25 (SNAP-25) and post-synaptic density protein-95 (PSD-95) are well established biochemical markers for pre-synaptic and post-synaptic fractions of spinal cord respectively [53,54]. Consistent with previous studies, Western blot analyses showed that SNAP-25 was highly expressed in pre-synaptic tissues, whereas PSD-95 was highly expressed in post-synaptic tissues (Figure 6A,B). Interestingly, in diabetic rats, α2-AR protein expression was significantly upregulated but only in pre-synaptic fractions of LSSC (Figure 6C); however, α2-AR protein expression remained unaltered in the post-synaptic fractions (Figure 6D).

Furthermore, dual-immunostaining with α2-AR and SNAP-25 (Figure 6E) or PSD-95 (Figure 6F) specific antibodies in LSSC revealed a significant overexpression and co-localization of these proteins in their respective pre-synaptic and post-synaptic tissues. Quantitation of the protein expression of α2-AR/SNAP-25 showed a significant elevation, whereas the α2-AR/PSD-95 remained unaltered as a function of diabetes (Figure 6G). As anticipated, chronic guanfacine administration mitigated diabetes-induced upregulation of pre-synaptic α2-AR in LSSC (Figure 6G). However, this effect of the drug was not seen in the post-synaptic fraction of the LSSC (Figure 6H).

## 4. Discussion

This study indicates that guanfacine has anti-hyperalgesic and anti-allodynic activities in a chronic rat model of T1DM-neuropathic pain, which were blocked by yohimbine. Rats with diabetes-induced hyperalgesia and allodynia had elevated levels of α2-AR mRNA and proteins in the pons, LSSC, and DRG. Chronic treatment with guanfacine alleviated diabetes-induced hyperalgesia and allodynia but had no activity in control rats. Furthermore, colocalization of α2-AR with SNAP-25 and PSD-95 implies the existence of α2-AR in both pre-synaptic and post-synaptic tissues. Lastly, chronic treatment with guanfacine resulted in a reduction of α2-AR in pre-synaptic, but not post-synaptic fractions.

Neuropathic pain is one of the most common chronic complications that develops in approximately 20–30% of diabetic patients with neuropathy [7, 8, 9]. In humans, it is characterized by spontaneous pain sensations, hyperalgesia, or allodynia. Studies have shown that rodents develop hyperalgesia or allodynia, signs of neuropathic pain, as a function of diabetes [16,55,56,57,58]. To date, no study has characterized the changes in cold sensitivity as a function of chronic diabetes in Wistar rat species. The present study addressed this gap with behavioral assessment of cold allodynia as a function of chronic diabetes and the associated molecular mechanism. Diabetic rats were found to have significantly reduced response time to cold stimulation. Worth noting, cold allodynia has been demonstrated in other rat species, such as Sprague-Dawley, in an acute model of diabetic neuropathic pain [55].

Recently, α2-AR agonists have been shown to alleviate neuropathic pain in an acute rat model of spinal nerve ligation (SNL)-induced hyperalgesia and mechanical allodynia [59]. Furthermore, the administration of the α2-AR agonist, 3-(2-chloro-6-fluorobenzil)- imidazolinide-2,4-dione (PT-31) shows anti-hyperalgesic and anti-allodynic effects, but not sedation. In contrast, clonidine exhibits a sedative profile [59]. On the other hand, treatment with clonidine failed to induce changes in tail flick latencies in a chronic model of T1DM [22]. In the present study, we showed that administration of clonidine, in comparison to guanfacine, induced changes in reaction latencies to thermal (heat/cold) and withdrawal thresholds to mechanical stimuli acutely. However, the anti-hyperalgesic and anti-allodynic effects of clonidine diminished following chronic administration. On the other hand, the anti-hyperalgesic and anti-allodynic activities of guanfacine were superior to clonidine and were sustained even after a period of no drug treatment (Figure 3 and Appendix A).

Guanfacine is a selective α2-AR agonist, unlike moxonidine, dexmedetomidine, or clonidine, which possess dual selectivity toward α2-AR and imidazoline receptors [28]. No study has yet evaluated the effects of guanfacine in diabetes-induced neuropathic pain. However, treatment with guanfacine was shown to generate antinociception in formalin-induced visceral-pain rat model [29]. The present study revealed that guanfacine exhibits anti-hyperalgesic and anti-allodynic effects, and abrogates diabetes-induced thermal hyperalgesia, cold allodynia, and mechanical allodynia in a Wistar rat chronic diabetic model, but not in control rats. The pronounced stimulus in diabetic rats indicates that the noradrenergic system plays a key role in diabetic neuropathic pain, and the precise role of α2-AR was further supported by the observed phenotype where the anti-hyperalgesic and anti-allodynic activities of guanfacine were antagonized by the α2-AR antagonist, yohimbine. Results of the current study are, to some extent, similar to a previous study in which α2-AR transcripts were significantly elevated in DRG spinal nerve-ligated rats compared to sham-operated rats [1]. Nevertheless, no study has yet evaluated changes in the expression of α2-AR mRNA or protein levels as a function of chronic diabetes or in response to guanfacine administration, although, it has been well documented that uncontrolled diabetes in animal models causes neuronal degeneration and atrophy [60,61,62]. We observed similar phenotype in both LSSC and DRG tissues isolated from diabetic rats; however, chronic treatment with guanfacine improved tissue regeneration to levels comparable to that observed in control animals.

On the other hand, different observations were reported using rat models for other types of neuropathic pain. Studies have shown that spinal α2-AR immunoreactivity is reduced in an animal model for neuropathic pain, including chronic constriction injury of the sciatic nerve, complete sciatic nerve transection, and L5/L6 spinal nerve ligation [63]. Thus, the observed increase in α2-AR gene transcripts and protein levels in pons, LSSC, DRG, and particularly in the pre-synaptic tissues of diabetic rats, suggest these receptors may play an important role in neuropathic pain. Interestingly, chronic administration of guanfacine reduced α2-AR significantly and rescued the neuronal tissues phenotype. In accordance, a recent study revealed that chronic treatment with guanfacine increased NA release in the ventral tegmental area, orbitofrontal cortex, and reticular-thalamic nucleus through the downregulation of α2-AR in the locus coeruleus, ventral tegmental area, and orbitofrontal cortex [62]. Furthermore, treatment with guanfacine chronically enhanced noradrenergic neurotransmission by attenuating the inhibitory α2-AR function [64]. 

Pre-synaptic α2-AR are coupled with Gα_i_ proteins and function as auto-receptors. Overexpression of α2-AR may enhance Gα_i_ function, decrease adenylate cyclase activity, and inhibit cyclic adenosine monophosphate (cAMP) production and downstream intracellular signal transmission, which will eventually alter neuronal activity and NA release [65,66]. In the current study, as a prospective signaling mechanism, we studied the expression level of the neuronal-associated coupling G-protein, Gα_i_, using Western blotting analysis. As observed in Appendix A, the increased expression of pre-synaptic α2-AR in the LSSC of diabetic rats was associated with an increase in Gα_i_, indicating a reduction in NA release through the activation of the Gα_i_ mediated downstream signaling pathway. Furthermore, chronic treatment with guanfacine downregulated Gα_i_ in the LSSC of the diabetic rats, and may have downregulated the pre-synaptic α2-AR and diminished the autoinhibitory negative feedback mechanism. However, the acute effect of guanfacine in reducing pain transmission in T1DM but not in control rats may be related to its ability to inhibit diabetes-induced upregulation in the expression of key neuronal hyperalgesic molecules such as transient receptor potential vanilloid 1 (TRPV1) and/or transient receptor potential ankyrin 1 (TRPA1) in several regions associated with pain transmission. Indeed, preliminary data from our laboratory clearly indicate that as a function of diabetes, guanfacine suppresses the increase in mRNA expression of TRPV1 and TRPA1 in several regions of the central nervous system including pons, LSSC, and DRG (unpublished data).

## 5. Conclusions

In conclusion, the current study shows that the expression of α2-AR was altered as a function of diabetes and this abnormality was ameliorated in response to chronic administration of guanfacine. Furthermore, guanfacine also alleviated diabetes-induced thermal hyperalgesia, cold allodynia, and mechanical allodynia. These biochemical and behavioral effects of guanfacine, viewed in the context of diabetes, are elucidated schematically in Figure 7. Overall, the current data suggest a positive signal for extending these studies to randomized placebo control clinical trials aimed at examining whether guanfacine, an FDA-approved drug for hypertension and attention deficit hyperactivity disorder, is able to ameliorate diabetic neuropathic pain in human subjects.

## 6. Limitations of the Study

In the current study, we used STZ to selectively destroy pancreatic β-cells in rats and generate an animal model of T1DM. This easily reproducible model of T1DM has been shown to develop key chronic diabetic complications such as neuropathy [67], nephropathy [68], atherosclerosis [69], impaired wound healing [70], and neurochemical imbalances within the central nervous system [17,71]. However, because STZ may be toxic to tissues and organs other than pancreatic β-cells, such a model is less likely to mimic human T1DM. This is particularly true when using a single dose of STZ (55 mg/Kg), as was the case in our study. Indeed, a high single dose of STZ (50–70 mg/Kg) directly and rapidly destroys pancreatic β-cells, and the resulting effect lacks some features of T1DM, such as pancreatic insulitis [72]. To this end, there is ongoing research in our laboratory on modifications of the existing model that will more adequately reflect some key features of human T1DM.

The second limitation of this study is that presynaptic and postsynaptic expression of α2-AR in response to guanfacine was confirmed using immunofluorescence and not a Western blot-based technique, which is the gold standard for determining protein expression in cells, tissues, and organs.

## Figures and Tables

**Figure 1 pharmaceutics-14-02146-f001:**
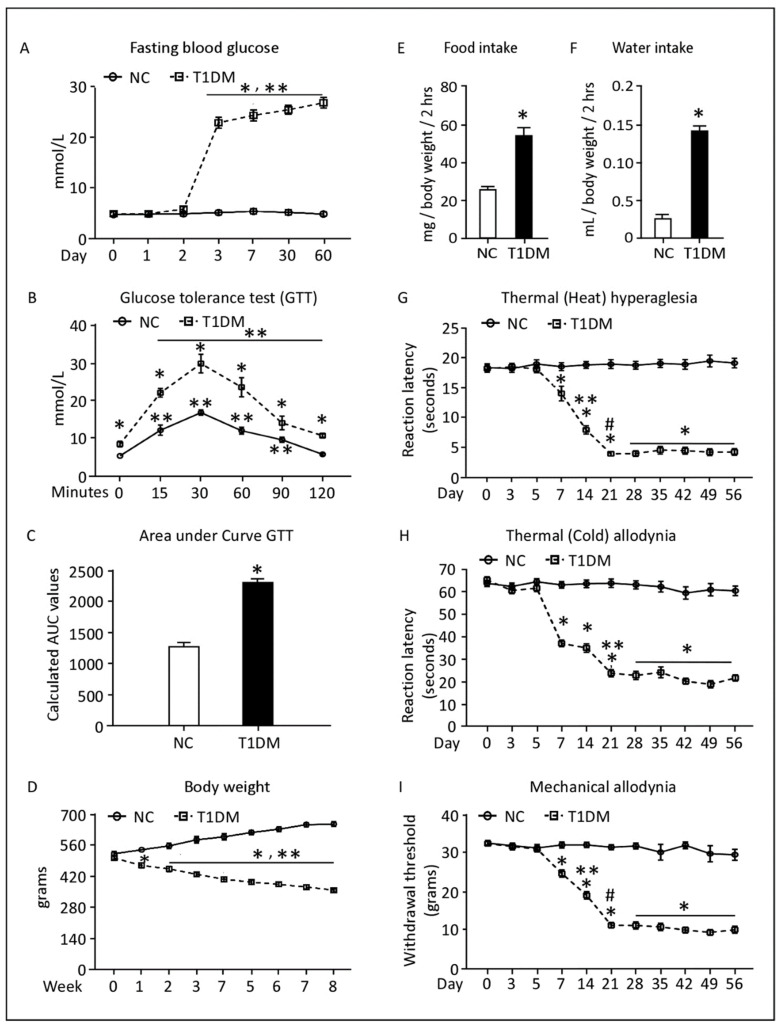
Biochemical, physical, and behavioral characteristics of control and T1DM rats. Time course of fasting blood glucose levels (**A**). Intraperitoneal glucose tolerance test and area under the curve (**B**,**C**), body weight, food consumption, water intake (**D**–**F**) and nociceptive behavior (**G**–**I**) in response to thermal (heat/cold) and mechanical stimuli. Data are expressed as means ± SEM of at least 12 animals/group. * Significantly different from corresponding control values at *p* ≤ 0.05. ** Significantly different from the corresponding first 48 h of STZ treatment values at *p* ≤ 0.05. # Significantly different from the corresponding first 14 days of STZ treatment values at *p* ≤ 0.05. Abbreviations: NC, Normal Control; T1DM, Type 1 Diabetes Mellitus; GTT, glucose tolerance test.

**Figure 2 pharmaceutics-14-02146-f002:**
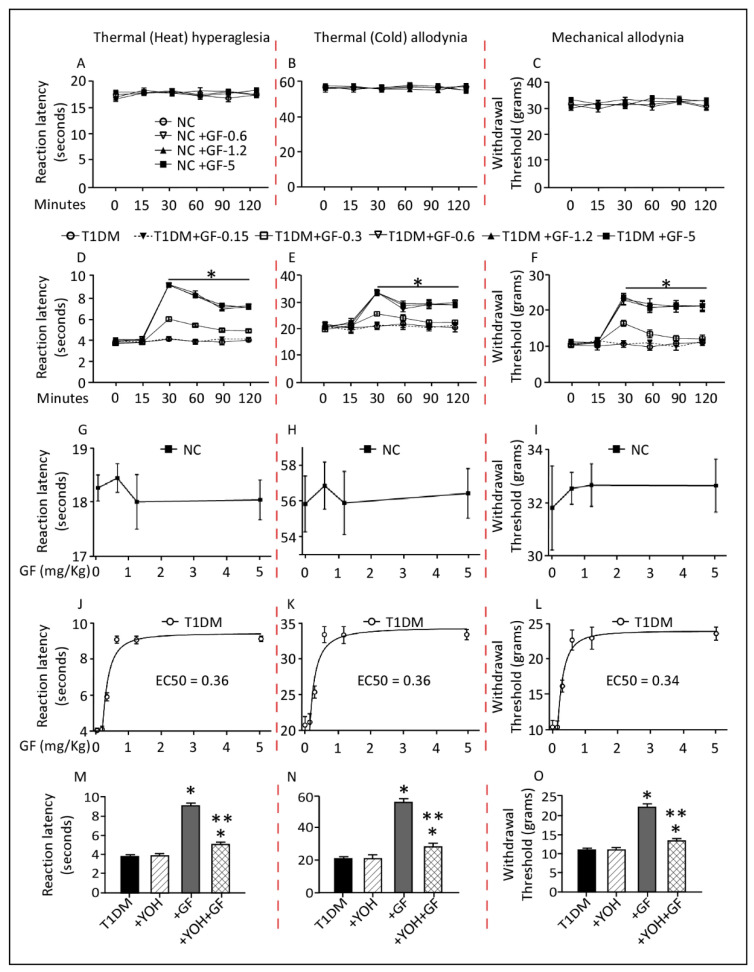
Time and dose response curves of administered guanfacine in control and diabetic rats. Guanfacine dose-dependently elevated thermal (heat/cold) and mechanical threshold (**D**–**F**) in diabetic but not in control (**A**–**C**) rats. Guanfacine in a concentration-dependent manner elevated pain threshold in diabetic (**J**–**L**) but not in control (**G**–**I**) rats at the 30 min time-point following administration. Yohimbine antagonizes guanfacine’s analgesic effects (**M**–**O**). Data are expressed as means ± SEM obtained from five animals/group. * Significantly different from corresponding diabetic values at *p* ≤ 0.05. ** Significantly different from corresponding guanfacine values at *p* ≤ 0.05. Abbreviations: NC, Normal Control; T1DM, Type 1 Diabetes Mellitus; YOH, yohimbine; GF, guanfacine; EC, effective concentration.

**Figure 3 pharmaceutics-14-02146-f003:**
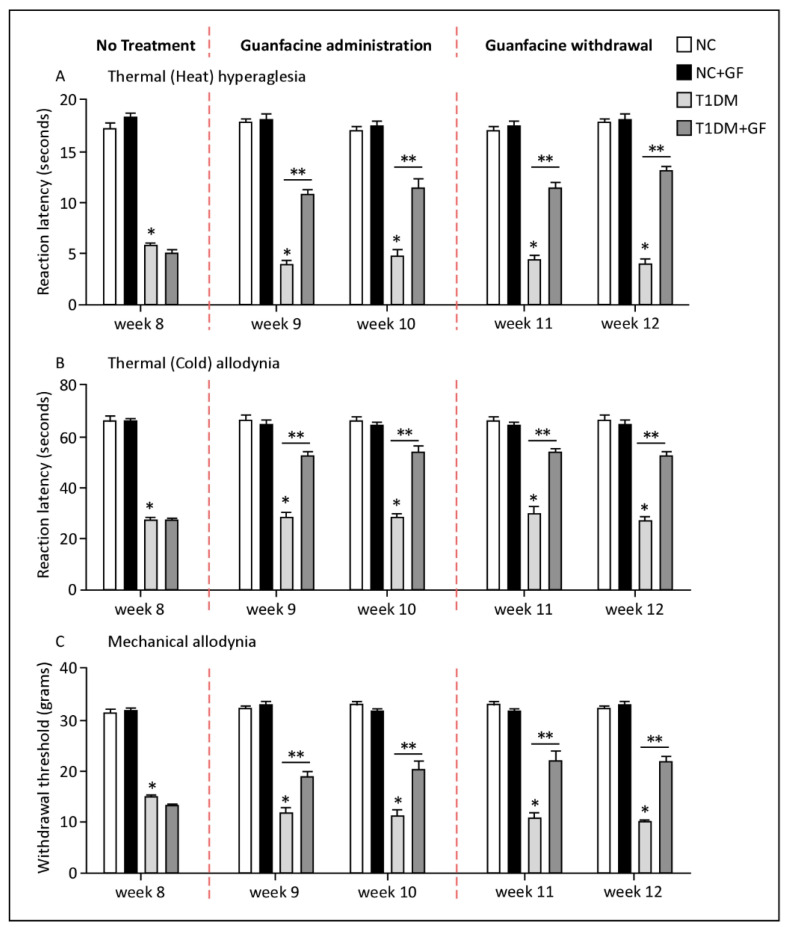
Persistent guanfacine analgesic effects following two weeks of withdrawal. Guanfacine (0.6 mg/Kg) was administered intraperitoneally daily for a period of two weeks. Pain transmission tests were performed at weekly interval for two weeks of guanfacine administration followed by two weeks of drug withdrawal. Data are expressed as means ± SEM obtained from five animals/group. * Significantly different from corresponding control values at *p* ≤ 0.05. ** Significantly different from corresponding diabetic values at *p* ≤ 0.05. Abbreviations: NC, Normal Control; T1DM, Type 1 Diabetes Mellitus; and GF, guanfacine.

**Figure 4 pharmaceutics-14-02146-f004:**
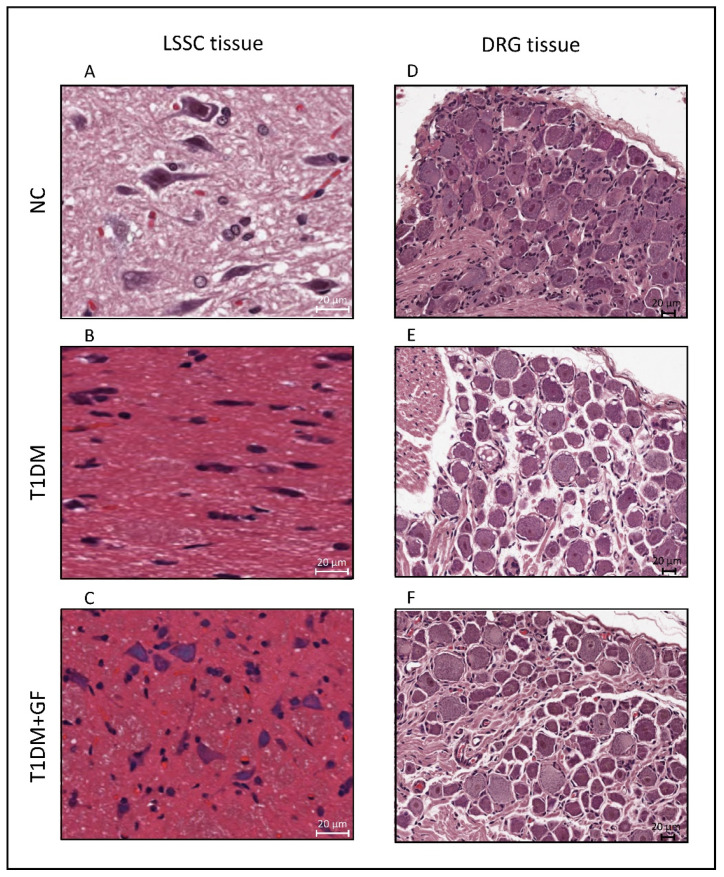
Chronically administered guanfacine halts diabetes-induced neuronal degeneration. Guanfacine (0.6 mg/Kg) was administered intraperitoneally daily for a period of two weeks. Hematoxylin and Eosin staining of LSSC (**A**–**C**) and DRG (**D**–**F**), respectively (magnification: 40× and scale bar: 20 µm). Abbreviations: NC, Normal Control; T1DM, Type 1 Diabetes Mellitus; GF, guanfacine; LSSC, lumbar segment of the spinal cord; and DRG, dorsal root ganglia.

**Figure 5 pharmaceutics-14-02146-f005:**
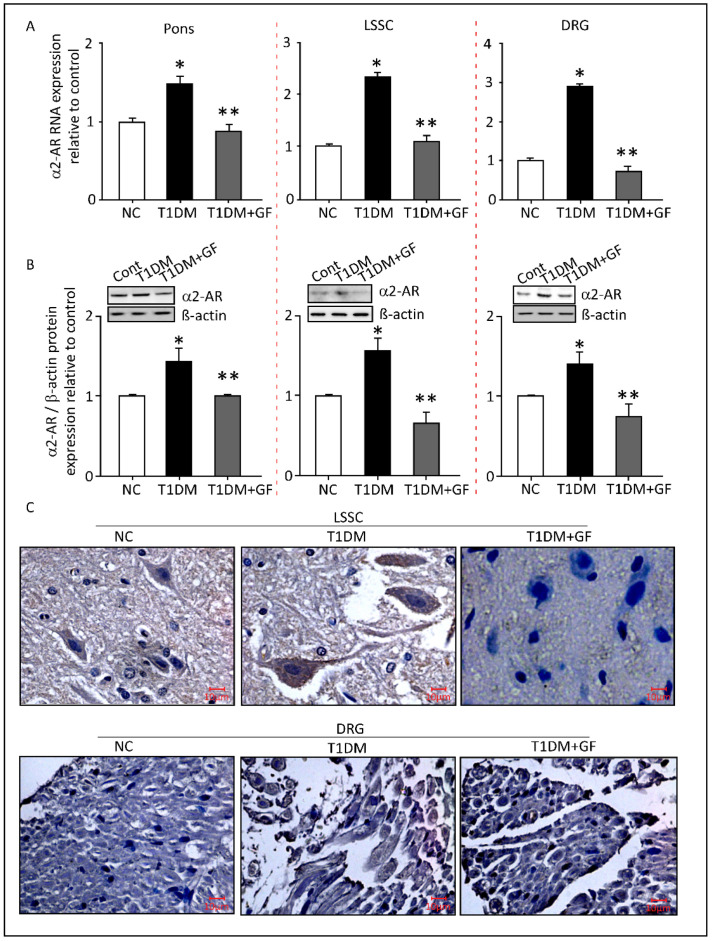
Chronically administered guanfacine reverses diabetes-induced upregulation of neuronal α2-AR. Expression of α2-AR was quantified at mRNA and protein levels using qRT-PCR, Western blotting, and immunohistochemistry-based technique, respectively. Guanfacine (0.6 mg/Kg) was administered intraperitoneally daily for a period of two weeks. Relative α2-AR mRNA expression (**A**). Protein bands imaged with the ChemiDoc MP were analyzed using Image Lab software (**B**). Representative immunohistochemical staining images for the expression of α2-AR in the LSSC and DRG tissue sections (**C**). Brown staining indicates neurons with positive expression (magnification: 40× and scale bar: 10 µm). Data are expressed as the mean ± SEM of values obtained from five animals/group. * Significantly different from corresponding control values at *p* ≤ 0.05. ** Significantly different from corresponding diabetic values at *p* ≤ 0.05. Abbreviations: NC, Normal Control; T1DM, Type 1 Diabetes Mellitus; GF, guanfacine; α2-AR, alpha-2A adrenoceptors; LSSC, lumbar segment of the spinal cord; and DRG, dorsal root ganglia.

**Figure 6 pharmaceutics-14-02146-f006:**
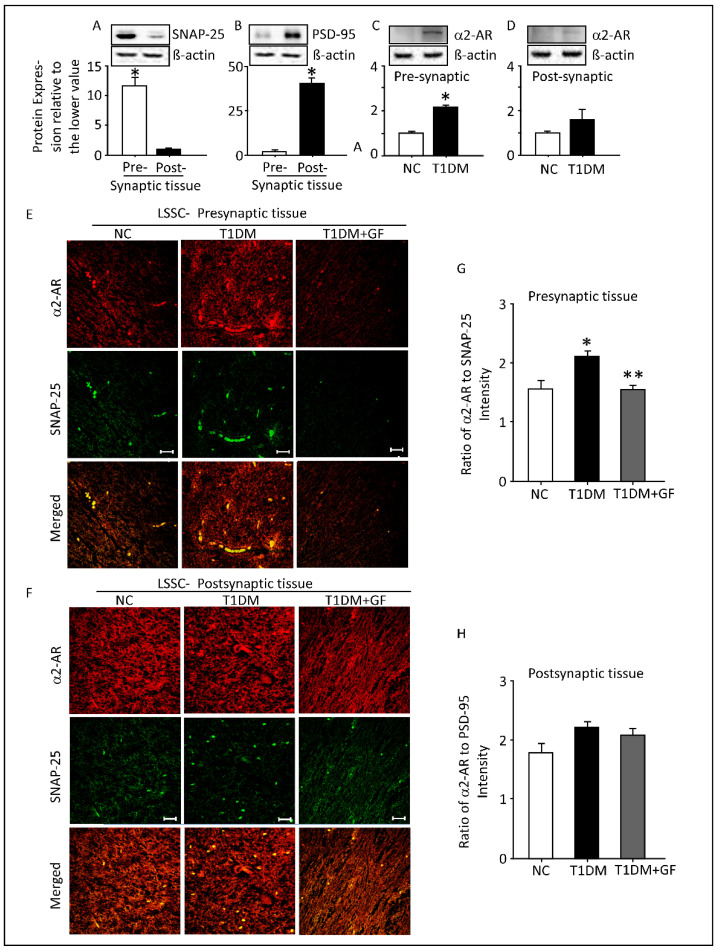
Expression of α2-AR in the pre-synaptic and post-synaptic fractions of the spinal cord lumbar region. The pre-/post-synaptic fractionation protocol was verified by Western blot using antibodies directed against the pre-synaptic SNAP-25 (**A**) and post-synaptic PSD-95 (**B**) protein marker. Diabetes upregulates pre-synaptic (**C**) but not post-synaptic (**D**) spinal α2-AR proteins. Immunofluorescence localization of pre-synaptic (**E**) and pos-synaptic (**F**) α2-AR in control, diabetic and diabetic treated with guanfacine for two weeks. Fluorescence intensity of α2-AR relative to SNAP-25 (**G**) and PSD-95 (**H**). Protein bands imaged with the ChemiDoc^TM^ MP were analyzed using Image Lab software. Each bar represents the mean ± SEM of values obtained from five animals/group. * Significantly different from corresponding or control values at *p* ≤ 0.05. ** Significantly different from corresponding diabetic values at *p* ≤ 0.05. Representative confocal images at magnification: 40X and scale bar: 20 µm. Abbreviations: NC, normal control; T1DM, Type 1 Diabetes Mellitus; GF, guanfacine; α2-AR, alpha-2A adrenoceptors; LSSC, lumbar segment of the spinal cord; SNAP-25, synaptosomal-associated protein-25; and PSD-95, postsynaptic density-95.

**Figure 7 pharmaceutics-14-02146-f007:**
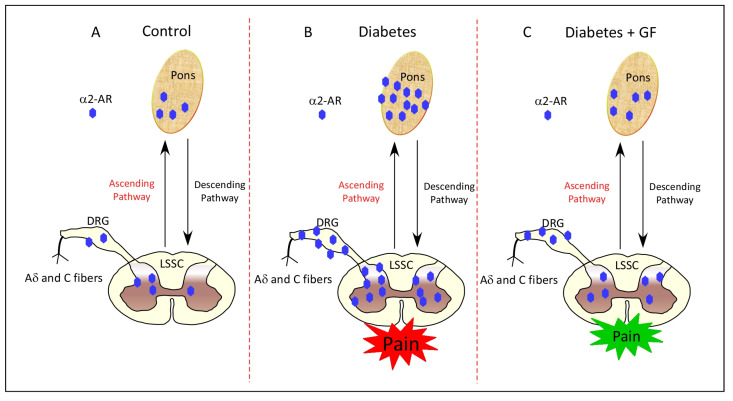
Schematic representation of presynaptic α2-AR as a function of diabetes, neuropathic pain and guanfacine treatment. In comparison with the control status (**A**), diabetes augments neuronal expression of presynaptic α2-AR in the pons, LSSC and DRG (**B**). Guanfacine administered chronically for two weeks reversed diabetes-induced upregulation of neuronal presynaptic α2-AR (**C**). Arrows indicate the ascending/descending pain transmission pathways. Abbreviations: GF, guanfacine; α2-AR, alpha-2A adrenoceptors; LSSC, lumbar segment of the spinal cord; and DRG, dorsal root ganglia.

**Table 1 pharmaceutics-14-02146-t001:** Antibodies used in the study.

Antibody (Vendor)	Dilution
Alpha-2A adrenoceptor (α2-ARs) (Abcam, Cambridge, MA, USA)	1:500, 1:100
SNAP-25 (Santa Cruz Biotechnology, Inc., Dallas, TX, USA)	1:1000, 1:100
PSD-95 (Novus Biologicals, Centennial, CO, USA)	1:2000, 1:100
G-α_i_ (Santa Cruz Biotechnology, Inc., Dallas, TX, USA)	1:1000
Secondary antibody, horseradish peroxidase (HRP) anti-rabbit IgG (Cell Signaling, Technology, Inc., Danvers, MA, USA)	1:3000
Secondary antibody, HRP-linked anti-mouse IgG (Cell Signaling Technology, Inc., Danvers, MA, USA)	1:3000
AlexaFluor-488-labeled goat anti-mouse IgG antibody (Invitrogen, Carlsbed, CA, USA)	1:100
AlexaFluor-546 labeled goat anti-rabbit IgG antibody (Invitrogen, Carlsbed, CA, USA)	1:100

## Data Availability

Data is contained within the article.

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
