# Peer review of "Guanfacine Normalizes the Overexpression of Presynaptic α-2A Adrenoceptor Signaling and Ameliorates Neuropathic Pain in a Chronic Animal Model of Type 1 Diabetes"

_pharmaceutics, 2022, doi:10.3390/pharmaceutics14102146_

Round 1
Reviewer 1 Report
Dear Editor,
Please find my comments for the manuscript below:
- OGTT/or ITT not done to confirm T1D. Such methods are gold standard method to confirm diabetes.
- Figure 2 (GHI) and its relevance not explained in the text.
- Where is Figure 2 (M-O)?
- Not convinced with Figure 5B, Actin expression is also increased in diabetic tissue. Quality of the western blot is also poor.
- Out of 50 diabetic rats, how many have complications? How many diabetic rats died during the study. Do all of them are otherwise healthy and showed no other symptoms of micro or macros vascular complications. Do you remove those rats from the study?
- Figure 6D, over interpretation of results- alpha -2AR protein not significantly increased still the term ‘up-regulated’ was used.
- SNAP-25 was expressed more in pre-synaptic tissues whereas PSD-95 was higher in post-synaptic. Is that normal? Why not these proteins were evaluated using western in guanfacine treatment condition. Interpretation of these results were very confusing.
- In general, the results and figures (and the text within figures) were not presented robustly in the manuscript. Also, standard abbreviations must be used in the figures wherever required to differentiate between control and diabetic (T1D) animals. The current manuscript also lacks accurate interpretation of results.
Author Response
Our responses to the excellent comments presented by reviewer 1 are itemized below.
- OGTT/or ITT not done to confirm T1D. Such methods are gold standard method to confirm diabetes.
As per the reviewer’s request, the data regarding glucose tolerance test (GTT) were included in Figure 1B and C; and detailed in the ‘Material and Methods’ section (please see Page 4, Lines 132-137) and in the ‘Results’ section (please see Pages 8-9, Lines 316-322).
- Figure 2 (GHI) and its relevance not explained in the text.
We thank the reviewer for his/her comment. Data representing pain tests in normal control were included in Figure 2 G-I and a detailed account regarding these figures was included in the figure legend (please see Page 12, Lines 375-377) and in the result section (please see Page 11, Lines 357-362).
- Where is Figure 2 (M-O)?
We thank the reviewer for his/her kind comment. We revised and corrected Figure 2. These essential data are now included in Figure 2 (M-O) and they are also detailed in the Result section (please see Page 11, Lines 363-371).
- Not convinced with Figure 5B, Actin expression is also increased in diabetic tissue. Quality of the western blot is also poor.
As per reviewer’s comment, we included a new representative image of β-actin and α2-AR protein expression in LSSC (please see Page 16, Figure 5B). As previously described, the experiments were performed several times, at least three times with n=5, and the mean and SEM were determined.
- Out of 50 diabetic rats, how many have complications? How many diabetic rats died during the study. Do all of them are otherwise healthy and showed no other symptoms of micro or macros vascular complications. Do you remove those rats from the study?
We thank the reviewer for his/her comment. Rats with severe diabetic symptoms representing about 9%, stemming from ketoacidosis, such as blindness, anorexia, slow movement, were excluded from the study. The starting number of animals was 55. The relatively healthy 50 animals were used in our study.
- Figure 6D, over interpretation of results- alpha-2AR protein not significantly increased still the term ‘up-regulated’ was used.
We thank the reviewer for his/her comment. The term upregulation was used to describe the overexpression of presynaptic alpha-2AR which was significantly different from corresponding control values (please see Figure 6C). However, it is noteworthy that the expression of postsynaptic alpha-2AR remains unchanged (please see Figure 6D). This essential notion was included in the result section (please see Page 17, Lines 466-469).
- SNAP-25 was expressed more in pre-synaptic tissues whereas PSD-95 was higher in post-synaptic. Is that normal? Why not these proteins were evaluated using western in guanfacine treatment condition. Interpretation of these results were very confusing.
Yes, it is normal since SNAP-25 is a well-established biochemical marker for pre-synaptic tissues whereas PSD-95 is a marker for post-synaptic tissues. This notion was detailed in the text of this manuscript with the appropriate references (please see Page 17, Lines 462-464).
Western blot-based technique was not used to determine the expression of presynaptic and postsynaptic of alpha2-AR in guanfacine treatment condition and we believe that this was one of the limitations of our study; this notion was included in section 6 under the subtitle ‘Limitations of the study’’ (please see Page 21, Lines 594-610).
We have revised the interpretation of our results incorporating most of the referees’ comments and we feel now that the results become much clearer to the reader.
- In general, the results and figures (and the text within figures) were not presented robustly in the manuscript. Also, standard abbreviations must be used in the figures wherever required to differentiate between control and diabetic (T1D) animals. The current manuscript also lacks accurate interpretation of results.
We believe that incorporating the 3 referees’ valuable comments into our manuscript has strengthened the manuscript and improved the interpretation of the results presented. As per reviewers’ comments, some figures and paragraphs of the text were modified to clearly indicate the changes that occur as a function of diabetes.
We also used standard abbreviations throughout the manuscript, including figures and legends: T1DM = Type-1 Diabetes Mellitus, NC = normal control, as well as the Materials and Methods (please see Page 4, Lines 151-157).

Reviewer 2 Report
The authors submitted a research article in which they investigated the hypothesis that the development of neuropathic pain stems from a defect in the central noradrenergic system activity in T1DM and chronic administration of guanfacine restores the defect of the noradrenergic system and ameliorates neuropathic pain. They used streptozotocin -induced diabetic rats as a model of the disease and randomly divided into vehicle (20 rats, non-diabetic) and treatment (50 rats, diabetic) groups. The acute and chronic effects of guanfacine was evaluated at escalating doses. The authors finally established that guanfacine had anti-hyperalgesic and anti-allodynic activities in a chronic rat model of T1DM-neuropathic pain, which were blocked by yohimbine. Therefore, they suggested that guanfacine could downregulated Gαi in the LSSC of the diabetic rats, thus, it may have downregulated the pre-synaptic α2-AR and diminished the autoinhibitory negative feedback mechanism. The aim of the study is clear and concise. The manuscript has a logical structure and is well referenced. The subsections of the paper are well-balanced and have labible tables and figures. The section "Discussion" covers all aspects of the study. The conclusive part semms to be informative and attractive for readers. Overall, I congratulate the authors on the study, although I would like to put forward several issues to discuss.
1. Neuropathic pain in humans has several molecular mechanisms, such as NETs-related ischemia, which cannot be executed in the streptozotocin model of diabetes. However, the majority of clinically significant changes have been evaluated in the study. The authors should add short comment in the section "Study limitations" to clearly articulate limitations of the models.
2. The authors should add their opinion about clinical perspective to use the findings taking into consideration other aspects of pharmacodynamics of guanfacine
Author Response
Our responses to the excellent comments presented by reviewer 2 are itemized below.
- Neuropathic pain in humans has several molecular mechanisms, such as NETs-related ischemia, which cannot be executed in the streptozotocin model of diabetes. However, the majority of clinically significant changes have been evaluated in the study. The authors should add short comment in the section "Study limitations" to clearly articulate limitations of the models.
As per referee’s suggestion, a new section was added to the manuscript under the subtitle of “Limitations of the study’’ (please see Page 21, Lines 594-610).
- The authors should add their opinion about clinical perspective to use the findings taking into consideration other aspects of pharmacodynamics of guanfacine
As per referee’s suggestion, this essential notion was added in the conclusion section of the manuscript (please see Page 21, Lines 583-592).

Reviewer 3 Report
This manuscript shows the results of experiments in a rat streptozotocin-induced diabetes model, investigating neuropathic pain. The investigators show that use of an alpha-2A receptor agonist, guanfacine, acutely and chronically reduces behavioral measures of neuropathic pain. They also find that chronic guanfacine treatment reduces presynaptic expression of the alpha-2A receptor. Interestingly, although diabetes leads to neurodegenerative changes in several regions examined, chronic guanfacine treatment reduces these changes. The manuscript is well written, and the experimental design is adequate to the question. The manuscript would benefit from a couple minor clarifications, and some addition to the discussion.
- The discussion falls short in explaining possible mechanisms for the guanfacine effect. In particular, the changes seen in nociception are similar in acute treatment (ie as shown in Fig. 2) and in chronic treatment (Fig. 3). Presumably, the time frame for anti-nociceptive effects in Figure 2 are too early for significant changes in expression of alpha-2A receptor. Since alpha-2 receptors function as autoreceptors which reduce norepinephrine release, acute activation of these receptors would have a different effect on norepinephrine signaling than chronic reduction in their expression. Acute activation would decrease norepinephrine release, while decreased expression of the receptor after chronic treatment would be expected to reduce the sensitivity of the receptor and thus lead to increased norepinephrine release. Thus, one might expect acute administration of guanfacine to have a different effect on nociception than chronic administration. Another possible explanation might be that diabetes reduces norepinephrine signaling, leading to decreased expression/increased sensitivity of the autoreceptor, which is reduced by activating the autoreceptor using guanfacine.
- What do the authors think are potential mechanisms for the neuroprotective/restorative effects of guanfacine on histologic appearance of the neural tissue?
- In the conclusion, the authors state that "guanfacine may have abolished the auto-inhibitory negative feedback mechanism." I don't think they have addressed this sufficiently to make this conclusion. Presumably this is reduced, as a result of reduced expression of alpha2-A receptors, but likely not abolished.
- As a minor point, I suggest rewording the phrase “guanfacine restores diabetes-induced upregulation of alpha2-AR” (e.g. in the title for section 3.5). The wording obscures the meaning of the phrase, and could be fixed by replacing the word “restores” with “reverses” or something similar.
Author Response
Our responses to the excellent comments presented by reviewer 3 are itemized below.
1- The discussion falls short in explaining possible mechanisms for the guanfacine effect. In particular, the changes seen in nociception are similar in acute treatment (ie as shown in Fig. 2) and in chronic treatment (Fig. 3). Presumably, the time frame for anti-nociceptive effects in Figure 2 is too early for significant changes in expression of alpha-2A receptor. Since alpha-2 receptors function as auto-receptors which reduce norepinephrine release, acute activation of these receptors would have a different effect on norepinephrine signaling than chronic reduction in their expression. Acute activation would decrease norepinephrine release, while decreased expression of the receptor after chronic treatment would be expected to reduce the sensitivity of the receptor and thus lead to increased norepinephrine release. Thus, one might expect acute administration of guanfacine to have a different effect on nociception than chronic administration. Another possible explanation might be that diabetes reduces norepinephrine signaling, leading to decreased expression/increased sensitivity of the auto-receptor, which is reduced by activating the auto-receptor using guanfacine.
We would like to thank the reviewer of his/her insight and impact in improving our manuscript. We summarized the possible mechanism(s) for the acute and chronic antinociceptive effect of guanfacine (please see Pages 20-21, Lines 573-581).
2- What do the authors think are potential mechanisms for the neuroprotective/restorative effects of guanfacine on histologic appearance of the neural tissue?
We thank the reviewer for his/her comment. The neuroprotective effect of guanfacine might be very well related to its ability to suppress the heightened state of oxidative stress and to ameliorate the low-grade inflammation, both of these abnormalities were found by our laboratory to be associated with neuronal tissues of STZ diabetic rats (unpublished observations).
3- In the conclusion, the authors state that "guanfacine may have abolished the auto-inhibitory negative feedback mechanism." I don't think they have addressed this sufficiently to make this conclusion. Presumably this is reduced, as a result of reduced expression of alpha2-A receptors, but likely not abolished.
We agree with the comment of the referee, however the conclusion was modified.
4- As a minor point, I suggest rewording the phrase “guanfacine restores diabetes-induced upregulation of alpha2-AR” (e.g. in the title for section 3.5). The wording obscures the meaning of the phrase and could be fixed by replacing the word “restores” with “reverses” or something similar.
As per referee’s request, we have replaced the word “restore” with “reverse” throughout the manuscript (please see Page 2, Line 69; Page 3, Line 104; Page 14, Line 413; Page 15, Line 427; Page 17, Line 449; Page 25, Line 670).
